# Exploring the Therapeutic Potential of Ethyl 3-Hydroxybutyrate in Alleviating Skeletal Muscle Wasting in Cancer Cachexia

**DOI:** 10.3390/biom13091330

**Published:** 2023-08-30

**Authors:** Yu Zhou, Ruohan Lu, Fusheng Lin, Shu Chen, Qi-Qing He, Guoyang Wu, Caihua Huang, Donghai Lin

**Affiliations:** 1Key Laboratory for Chemical Biology of Fujian Province, MOE Key Laboratory of Spectrochemical Analysis and Instrumentation, College of Chemistry and Chemical Engineering, Xiamen University, Xiamen 361005, China; 20520200156095@stu.xmu.edu.cn (Y.Z.); 20520191151508@stu.xmu.edu.cn (R.L.);; 2Department of General Surgery, Zhongshan Hospital, Xiamen University, Xiamen 361005, China; linfusheng@xmzsh.com; 3Research and Communication Center of Exercise and Health, Xiamen University of Technology, Xiamen 361005, China; huangcaihua@xmut.edu.cn

**Keywords:** Ethyl 3-hydroxybutyrate, 3-hydroxybutyrate, cancer cachexia, ketogenic diet, metabonomics

## Abstract

Cachexia (CAC) is a debilitating metabolic syndrome. Although dietary interventions are attractive, long-term adherence to specific diets is difficult to maintain and can lead to systemic side effects. Ethyl 3-hydroxybutyrate (EHB) is a commonly used food additive found in wine and Tribolium castaneum. In this study, we investigated the effects of EHB administration in cachectic mice. After a single intraperitoneal injection of EHB into mice, 3-hydroxybutyrate (3-HB) levels were significantly increased in the serum and gastrocnemius of mice. The administration of EHB alleviated cachexia-related symptoms, ameliorated skeletal muscle atrophy, and improved survival in cachectic mice. In addition, the supplementation of cachectic mice with 3-HB by EHB administration significantly reduced tumor weights, indicating the anti-tumor effects of 3-HB. Remarkably, the addition of 3-HB to the culture medium significantly attenuated the C2C12 myotube atrophy induced by the culture supernatant of CT26 cell lines, highlighting its potential to counteract the destructive effects of tumor-derived elements on muscle tissue. NMR-based metabolomics analysis provided insights into the underlying mechanisms and revealed that the anti-cachexia effects of 3-HB treatment can be attributed to three key mechanisms: the promotion of the TCA cycle and the attenuation of proteolysis, the promotion of protein synthesis and the improvement of metabolic homeostasis, and a reduction in inflammation and an enhancement of the antioxidant capacity. This study provided compelling evidence for the protective effects of 3-HB treatment on the cachectic gastrocnemius and highlighted the efficacy of EHB administration as a ketone supplementation approach to achieve nutritional ketosis without the need for dietary restriction.

## 1. Introduction

Cachexia is a complex metabolic syndrome characterized by body weight loss and skeletal muscle wasting [1]. This syndrome is particularly prominent in severe chronic diseases and advanced-stage cancers, leading to disruptive metabolic imbalances, weight loss, fat depletion, and skeletal muscle atrophy [2,3,4]. Of particular concern is cancer cachexia (CAC), which accounts for 22–30% of cancer-related deaths and adversely affects treatment outcomes and quality of life [5,6,7].

The pathological process of CAC involves excessive inflammation and imbalanced energy expenditure, disrupting the delicate equilibrium between protein synthesis and proteolysis and contributing to skeletal muscle wasting [8,9]. Inflammatory factors, such as tumor necrosis factor alpha (TNF-α), interferon gamma (INF-γ), interleukin-1 (IL-1), and interleukin-6 (IL-6), which are released by immune or tumor cells, promote proteolysis in skeletal muscle and suppress protein synthesis through the intracellular signaling pathways [10,11,12]. The upregulation of the ubiquitin–proteasome pathway, indicated by increased expression of E3 ligases like the muscle atrophy F-box protein (atrogin-1) and muscle-specific ring finger protein 1 (MuRF-1), contributes to the degradation of essential structural components within muscle fibers [13,14]. Additionally, CAC is characterized by downregulated protein anabolic signaling pathways, such as the PI3K-Akt pathway and mTOR-dependent protein synthesis [15].

Cachexia is associated with malnutrition and poses a significant challenge to conventional nutritional interventions [16,17]. A ketogenic diet emerged as a promising strategy for cancer treatment, characterized by increased fat intake and restricted carbohydrates [18,19,20]. In various physiological conditions such as a high-fat diet, carbohydrate restriction, starvation, and fasting, the body produces ketone bodies, including 3-hydroxybutyrate (3-HB), acetoacetate, and acetone [21]. While a ketogenic diet showed promise in cancer treatment [20,22], it is difficult to implement due to its high fat, low carbohydrate, and low protein content. A ketogenic diet involves limiting certain foods, which raised worries about its effect on cardiovascular health [23,24,25]. 3-HB is the primary bioactive component of a ketogenic diet, as well as being the major ketone body and the primary circulating ketone in the bloodstream [20]. The exogenous supplementation of 3-HB can be achieved through various methods, including the use of a ketogenic diet, ketone salts, or ketone esters [24]. However, the exogenous supplementation of 3-HB in an acid or salt form is not recommended due to the accompanying acid/salt load, which could lead to gastrointestinal distress and the possibly adverse consequences of cation overload or acidosis [25,26].

Reports showed that the ingestion of ketone esters, such as (R)-3-hydroxybutyl (R)-3-hydroxybutyrate, can rapidly raise blood ketone concentrations, mimicking the metabolic effects of a ketogenic diet [27,28] and show promise in surpassing the limitations of a ketogenic diet [29]. Ethyl 3-hydroxybutyrate (EHB) is a ketone ester, which was originally discovered as a flavoring compound in wine, that is commonly used as a food additive due to its pleasant aroma and fruity flavor, [29,30]. However, no relevant published study was reported to investigate the function of EHB in other areas. It is still unclear if EHB can hydrolyze and form 3-HB in the body, which could be effective at reducing cachectic muscle wasting. To decide the therapeutic potential of EHB for this purpose, it is essential to resolve this uncertainty. 

In this study, we aimed to assess the therapeutic potential of EHB administration in alleviating skeletal muscle atrophy in a mouse model of colon cancer cachexia. We utilized NMR spectroscopy to evaluate the effect of a single intraperitoneal injection of EHB on 3-HB levels in the serum and gastrocnemius of mice. We observed a significant increase in the serum and gastrocnemius’s 3-HB levels, despite the absence of intact EHB. This indicated that the administration of EHB may result in similar effects as those of direct 3-HB treatment. We further investigated the effects of 3-HB derived from EHB hydrolysis on CAC mice. Evidently, 3-HB treatment significantly reduced tumor weights and ameliorated cachectic muscle wasting. This study highlighted the therapeutic potential of 3-HB treatment in alleviating cachectic muscle atrophy and promoted the exploration of EHB administration as a convenient exogenous supplementation of 3-HB for managing cancer cachexia.

## 2. Materials and Methods

### 2.1. Cell Culture Experiments

Mouse colon cancer cells (CT26) were obtained from National Biomedical Cell Resource Bank (BMCR, Beijing, China). CT26 cells were cultured in DMEM (Hyclone, Logan, UT, USA) supplemented with 10% fetal bovine serum (FBS), 100 units/mL penicillin, and 100 µg/mL streptomycin. Cells were cultured in a constant temperature incubator with 5% CO_2_ at 37 °C. Cells were harvested by digestion with 0.25% trypsin-EDTA (Hyclone, USA) when the confluency reached 85–95%. C2C12 myoblasts were cultured in growth medium consisting of DMEM supplemented with 10% fetal bovine serum in a 5% CO_2_-humidified atmosphere. For differentiation, subconfluent C2C12 was shifted from growth medium to differentiation medium composed of DMEM containing 2% horse serum for approximately 5 days. Conditioned medium (CM): CT26 cells were cultured in DMEM containing 10% fetal bovine serum (FBS) until 80% confluence. The medium was then replaced with serum-free DMEM for 48 h. The culture medium, which became CM from the secretome of tumor cells, was centrifuged at 13,000× *g* for 10 min to remove debris and immediately stocked at −80 °C until used. CM-induced C2C12 myotube atrophy: as previously reported [31], CM was diluted with 20% final differentiation medium to induce myotube atrophy. 3-hydroxybutyrate (at a final concentration of 20 mM) was added to 20% CM and maintained throughout the cell experiment.

### 2.2. Animal Experiments

The experimental animal protocol was approved by the Ethics Review Committee of Xiamen University, China, with the ethics of animal experimentation approval code XMULAC20200150. All relevant institutional and governmental guidelines and regulations governing the ethical treatment of animals were strictly adhered to throughout the study. BALB/c male mice, aged 6–8 weeks, were obtained from Shanghai SLAC Animal Company and housed in a controlled environment with a 12 h light/12 h dark cycle to ensure their well-being and minimize stress.

To evaluate the kinetic profiles of 3-HB levels in the bloodstream and skeletal muscle of mice, we collected serum and gastrocnemius samples from a total of 40 male mice. These mice received a single intraperitoneal injection of EHB at a dose of 300 mg/kg body weight. Sampling was performed at specific time points, including 0, 5, 10, 30, 60, 120, 240, and 480 min after EHB administration. At each time point, samples were obtained from 5 mice.

To explore the effects of EHB supplementation on cachectic muscle atrophy, we established a mouse model of colon cancer cachexia. The mice were divided into CAC-K, CAC, and NOR groups. After one week of acclimation to the housing conditions, CT26 cells (1 × 10^6^/100 µL) were subcutaneously injected into the right flank of the mice [32]. This injection induced cachexia-related symptoms, including significant body weight loss, muscle wasting, fat loss, emaciation, and reduced activity. A total of 30 mice exhibiting these symptoms were identified as cancer cachexia mice, comprising the CAC-K group (n = 19) and CAC group (n = 11). For comparison, a separate set of mice served as normal controls (NOR; n = 13) and received intraperitoneal injections of an equivalent volume of PBS. The CAC-K group of mice received intraperitoneal injections of EHB at a dose of 300 mg/kg/day determined in a pre-experiment, while the CAC group of mice received intraperitoneal injections of an equivalent volume of PBS. Similarly, the NOR group of mice were injected with PBS. Throughout the animal experiment, the food intake and body weight of the mice were monitored. On day 28, these mice were sacrificed, and tumor tissues, gastrocnemius muscles, and blood samples were collected to analyze the effects of EHB administration on colon cancer cachexia.

### 2.3. Histology Analysis

The mouse gastrocnemius muscles were fixed in 4% paraformaldehyde (PFA) to maintain their shape, then dehydrated with ethanol, and embedded in paraffin, before being cut into thin slices. The hematoxylin–eosin (H&E) staining technique was used to visualize the tissue morphology and identify any alterations in the gastrocnemius muscles. Image J software (V1.8.0) was used to measure the area of muscle per image, providing an objective measure of myofiber size. This quantification helps to evaluate any changes in the muscle tissue, such as myofiber atrophy or hypertrophy.

### 2.4. Western Blot Analysis

Western blotting was performed in accordance with a previously established protocol [33]. The protein concentration was measured using the BCA protein assay kit (Beyotime Biotechnology, Shanghai, China). Proteins were separated by 10–15% SDS-PAGE and then transferred to PVDF membranes (GE Healthcare, Shanghai, China) using a wet Trans–Bolt system. After blocking with 5% BSA (Beyotime Biotechnology), the membranes were incubated with specific primary antibodies at 4 °C, including anti-AKT (10176-2-AP, Proteintech, Wuhan, China), anti-phospho-AKT (AP3434A, ABGENT, San Diego, USA), anti-muscle-specific ring finger protein 1 (MP3401, ECM bioscience), and anti-glyceraldehyde-3-phosphate dehydrogenase (10494-1-AP, Proteintech). The corresponding secondary antibody conjugated horseradish peroxidase (Cell Signaling Technology, Danvers, MA, USA) was then applied to the membrane for 1 h at room temperature. For signal detection, both the enhanced chemiluminescence reagent (ECL, Beyotime Biotechnology) and ChemiScope Capture and Analysis system (ChemiScope 6000, CLiNX, Shanghai, China) were used.

### 2.5. Detection of Inflammatory Factors

Mouse orbital blood was collected and stored in a coagulation tube. Serum was obtained by centrifuging the blood sample (1000 RCF, 10 min, and 4 °C). The serum levels of inflammatory factors, such as Interleukin-1 (IL-1), Interleukin-6 (IL-6), Transforming Growth Factor-β (TGF-β), Interferon-γ (IFN-γ), Tumor Necrosis Factor-α (TNF-α), and toll-like receptor 4 (TLR-4), were measured by ELISA kit (SenBeiJia Biological Technology Co., Ltd., Nanjing, China). The sample was mixed with solid-phase purified antibody and HRP enzyme-labeled antibody to form an antibody–antigen–enzyme-labeled antibody complex. Color rendering was achieved by adding substrate TMB, which eventually turns yellow. The absorbance (OD) value of the sample was measured at 450 nm using a microplate reader. The serum concentrations of inflammatory factors were calculated based on a standard curve.

### 2.6. Measurement of Total Antioxidant Capacity

An ABST kit (Beyotime Biotechnology, Shanghai, China) was used to analyze the total antioxidant capacity of the gastrocnemius muscle. This kit works by oxidizing ABST to ABST+, and the antioxidant can hinder the formation of ABST+. The absorbance of ABST+ at 734 nm or 405 nm was then measured with a microplate reader.

### 2.7. Preparation of NMR Samples

Gastrocnemius samples were thawed in the refrigerator, and the membrane on the surface of the calf muscles and Achilles tendons were removed. A mixture of methanol, chloroform, and water (4:4:2.85), which was homogenized at 65 Hz for 60 s, was used to extract aqueous metabolites from the skeletal muscles. After 2 min of vortexing and 15 min of centrifugating (12,000 rpm and 4 °C), the upper water phase was taken, and the methanol was removed by nitrogen blowing. The solution was then lyophilized, and the powder samples were redissolved in NMR buffer (10% D2O, 50 mM PBS, and 1 mM 3-(trimethylsilyl) propionate-2, 2, 3, 3-d4 (TSP)) and transferred into 5 mm NMR tubes for NMR experiments. This protocol was based on the previously described method [33,34].

### 2.8. NMR Experiments

A Bruker AVANCE III HD 850 MHz NMR spectrometer (Bruker BioSpin, ettlingen, Germany) was used to acquire one-dimensional (1D) ^1^H-NMR spectra at 25 °C using the NOESYGPPR1D pulse sequence, with a relaxation delay of 4 s, 32 scans, and a spectral width of 20 ppm. Resonance assignments of metabolites were performed by a combination of the Chenomx NMR Suite (version 8.3, Chenomx Inc., Edmonton, Canada), Human Metabolome Database (HMDB, http://www.hmdb.ca/ (accessed on 1 September 2022)), and relevant published sources. To confirm these assignments, several two-dimensional (2D) NMR spectra were acquired, including 1H-1H total correlation spectroscopy (TOCSY) and 1H-13C heteronuclear single quantum correlation (HSQC).

### 2.9. Metabolomics Analysis

Multivariate statistical analysis of the NMR dataset was performed using SIMCA-P+ 14.0 software (MKS Umetrics, Malmo, Sweden). Unsupervised principal component analysis (PCA) and supervised partial least squares-discriminant analysis (PLS-DA) were used to analyze the metabolic profiles of the gastrocnemius muscles. To ensure the reliability of the PLS-DA model, a response permutation test with 200 cycles was conducted to assess its robustness. For metabolic pathway analysis, the MetaboAnalyst 5.0 webserver (http://www.MetaboAnalyst.ca (accessed on 1 September 2022)) was utilized [35]. Pathway topological analysis (PTA) was implemented to calculate the pathway impact value (PIV) for each metabolic pathway. Significantly altered metabolic pathways, referred to as significant pathways, were identified based on pairwise comparisons between CAC and NOR gastrocnemii, as well as between CAC-K and CAC gastrocnemii. The criteria for screening significant pathways were set at *p* < 0.05 and PIV > 0.1.

### 2.10. Statistical Analysis

The experimental data were presented as mean ± SD. Statistical analysis was conducted using IBM SPSS Statistics 22.0 software (IBM, New York, NY, USA). Pairwise comparisons among the three groups of mouse gastrocnemii were performed using one-way ANOVA followed by Tukey’s multiple comparison test. Statistical significance was represented as follows: *p* > 0.05 (not significant, ns), *p* < 0.05 (*), *p* < 0.01 (**), *p* < 0.001 (***), and *p* < 0.0001 (****).

## 3. Results

### 3.1. EHB Administration Elevated Serum and Gastrocnemius Levels of 3-HB in Mice

Our results revealed a significant increase in serum 3-HB levels, reaching a peak 5 min after EHB administration, with a 5.5-fold elevation compared to baseline levels (Figure 1A). Within 30 min of EHB administration, serum 3-HB levels returned to the baseline levels measured at 0 min after EHB administration. Similarly, we observed a remarkable 10-fold increase in the gastrocnemius 3-HB levels 10 min after EHB administration. However, no signal of EHB was present over time in the serum and gastrocnemius in normal mice, following a single intraperitoneal injection of EHB (Figure 1A,B). These results demonstrated the effective elevation of 3-HB levels in both the systemic circulation and skeletal muscle following a single intraperitoneal injection of EHB. Therefore, it was expected that the administration of EHB would have similar effects as direct 3-HB treatment.

### 3.2. EHB Administration Mitigated Cachexia-Related Symptoms in CAC Mice

We established a mouse model of colon cancer cachexia to investigate the effects of EHB administration on cachexia-related symptoms. As expected, the mouse model exhibited prominent cachectic features, including anorexia (Figure 2A), tumor-free body weight loss exceeding 5% (Figure 2B), and reductions in both fat and muscle mass (Figure 2C,D). CAC mice that received EHB administration displayed improvements in cachectic features (Figure 2). These mice demonstrated decreased losses of body weight and fat, decreased skeletal muscle wasting (Figure 2B–D), reduced tumor weights (Figure 2E), and increased survival rates (Figure 2F). This suggests that EHB administration is effective in alleviating cachectic features and may even have an anti-tumor effect. Although EHB administration significantly improved cachexia-related symptoms, it did not completely reverse the cachexia-induced loss of body weight, muscle weight, and fat weight, as shown by the statistically significant differences between CAC-K and NOR mice (Figure 2B–D).

### 3.3. EHB Administration Reduced Inflammation and Enhanced Total Antioxidant Capacity in CAC Mice

Chronic inflammation is a known contributor to skeletal muscle atrophy in cancer [10]. In line with this, CAC mice displayed markedly elevated serum levels of inflammatory factors, including IL-1, IL-6, TNF-α, TGF-β, IFN-γ, and the cachexia serum marker TLR-4. Significantly, EHB administration reduced the serum levels of IFN-γ and TGF-β, effectively restoring these inflammatory factors to normal levels (Figure 3A–F). It is worth noting that IFN-γ is associated with both oxidative stress in muscle and the promotion of pro-inflammatory cytokine synthesis [36]. Furthermore, histological analysis showed that EHB administration substantially improved the infiltration of inflammatory cells in the gastrocnemius muscle of CAC mice (Appendix A). These results demonstrated that EHB administration reduced inflammation by reducing the levels of inflammatory factors in cancer cachexia. Furthermore, we also measured the total antioxidant capacity (TAC) in the cachectic gastrocnemius to assess oxidative stress. CAC mice displayed a remarkably decreased TAC level, indicative of excessive oxidative stress in the cachectic skeletal muscle. Significantly, EHB administration almost fully recovered the cachexia-related decrease in the TAC level (Figure 3G).

### 3.4. EHB-Administration Ameliorated Skeletal Muscle Atrophy in CAC Mice

We performed hematoxylin–eosin staining on a histopathological section to observe structural change in the gastrocnemius. CAC mice exhibited distinctly decreased cross-sectional areas of muscle fibers, while EHB administration partially restored these cross-sectional areas (Figure 4A). In addition, we measured the expression of MuRF1 in the mouse gastrocnemius to assess muscle atrophy. CAC mice showed markedly upregulated expression of MuRF1, which was almost fully reversed by EHB administration (Figure 4B). Furthermore, CAC-K mice displayed an increased ratio of p-AKT/AKT, suggesting that EHB administration obviously activated the p-AKT/AKT signaling pathway inhibited by cancer cachexia, partly restoring the phosphorylation level of AKT (Figure 4B). These results indicated that EHB administration could obviously improve the unbalance between protein synthesis and proteolysis, significantly ameliorating the loss in skeletal muscle in CAC mice. EHB administration elevated the gastrocnemius levels of 3-HB in mice (Figure 1B). We also observed a significant alleviation of C2C12 myotubes atrophy induced by the culture supernatant of CT26 cell lines upon the addition of 3-HB in the culture medium, as evidenced by the expression levels of the atrophy-related protein MuRF1 (Appendix A). The CT26 Model group exhibited significantly elevated expression of MuRF1 compared to the Control group. However, in the CT26+3-HB group, the expression of MuRF1 was almost fully restored, indicating that supplementation with 3-HB could effectively reverse the expression of MuRF1 impaired by tumor cells. These results strongly suggested that the anti-cachectic effect of EHB was increasing the 3-HB content.

### 3.5. NMR-Based Metabonomic Analysis of Mouse Gastrocnemius

We recorded 1D 1H-NMR spectra on aqueous metabolites extracted from gastrocnemius muscle samples obtained from the NOR, CAC, and CAC-K groups (Appendix A). Based on the 1D 1H-NMR spectra, we assigned resonances to 34 metabolites (Appendix A). To validate the resonance assignments, we recorded 2D 1H-13C HSQC spectra to confirm the resonances of the identified metabolites (Appendix A). We performed principal component analysis (PCA) to gain insights into the metabolic profiles of the three groups of mouse gastrocnemius. The score plots of the PCA models illustrated the metabolic differences among NOR, CAC, and CAC-K mice, capturing the distinctions both between CAC and NOR mice and between CAC and CAC-K mice (Appendix A).

To further enhance the metabolic differences among the groups, we employed partial least squares-discriminant analysis (PLS-DA). This approach allowed us to maximize the metabolic separation between CAC and NOR mice, as well as between CAC-K and CAC mice, and identify the significant metabolites contributing to these metabolic distinctions. The score plots of the PLS-DA models clearly illustrated the distinct metabolic profiles of the cachectic gastrocnemius compared to those of the normal control, with noticeable alterations observed upon 3-HB treatment (Figure 5A,B). Additionally, the cross-validation plots obtained from response permutation tests validated the reliability of the PLS-DA models for the CAC vs. NOR and CAC-K vs. CAC comparisons (Appendix A). Based on the established PLS-DA models, we identified 13 significant metabolites from the CAC vs. NOR comparison and 16 significant metabolites from the CAC-K vs. CAC comparison using the criterion of VIP >1 (Figure 5C,D).

To quantify the relative levels of the metabolites in the gastrocnemius, we measured the relative integrals based on the weights of the gastrocnemius samples and the concentration of trimethylsilyl propionic acid (TSP). Subsequently, we performed a one-way analysis of variance (ANOVA), followed by Turkey’s multiple comparison test, to statistically compare the relative levels of metabolites among the NOR, CAC, and CAC-K groups. Using the criterion of *p* < 0.05, we identified 15 differential metabolites from the CAC vs. NOR comparison and 16 differential metabolites from the CAC-K vs. CAC comparison (Appendix A). This analysis allowed us to identify the important metabolites that exhibited significant alterations in their concentrations among the different groups, providing further insights into the metabolic changes associated with cancer cachexia.

Additionally, we identified the characteristic metabolites in the gastrocnemius using two criteria: VIP > 1 and *p* < 0.05. Specifically, we identified 12 characteristic metabolites from the CAC vs. NOR comparison and 13 characteristic metabolites from the CAC-K vs. CAC comparison. Interestingly, six characteristic metabolites were found to be common between these two comparisons, namely, valine, leucine, phenylalanine, fumarate, glutathione, and 2-methylglutarate (Figure 6).

Notably, in the CAC gastrocnemius, the levels of the three characteristic metabolites, valine, leucine, and phenylalanine, were increased compared to those of the NOR gastrocnemius. However, 3-HB treatment effectively decreased these levels (Figure 7). In contrast, the levels of the remaining three characteristic metabolites, fumarate, glutathione, and 2-methylglutarate, were significantly reduced in the CAC gastrocnemius compared to the NOR gastrocnemius. Interestingly, 3-HB treatment led to a profound increase in the levels of these metabolites (Figure 7). These results indicated that 3-HB treatment has the ability to significantly reverse the altered levels of these six characteristic metabolites, except for phenylalanine, in the CAC gastrocnemius, effectively restoring them to normal levels.

### 3.6. Pearson’s Correlations between Intergroup Variations in Serum Levels of Inflammatory Factors with Those in Gastrocnemius Levels of Metabolites

To compare the relative levels of metabolites among the NOR, CAC, and CAC-K groups of the gastrocnemius muscle, we constructed a heatmap plot that provides a visual representation of the variations in metabolite levels (Appendix A). Notably, multiple amino acids, such as alanine, valine, leucine, isoleucine, and proline, were increased in the CAC gastrocnemius, potentially reflecting enhanced protein degradation.

To further explore the relationship between the intergroup changes in the serum levels of inflammatory factors, expressions of catabolic and anabolic proteins, and the relative levels of the identified metabolites in the CAC and NOR groups of the mouse gastrocnemius, we conducted Pearson’s correlation analysis (Appendix A). The correlation analysis of CAC vs. NOR revealed several interesting findings. Inflammatory factors and the atrophy-related protein MuRF1 showed positive correlations with several metabolites, including lysine, phenylalanine, leucine, tyrosine, valine, inosine, 2-methylglutarate, isoleucine, and proline. Conversely, they showed negative correlations with ATP, 3-HB, glutathione, fumarate, glutamine, and pyruvate (Appendix A). Interestingly, the expression of phosphorylated p-AKT showed opposite correlations with these factors (Appendix A).

These correlation results provided valuable insights into the associations among inflammatory factors, protein expression, and metabolic changes in the CAC gastrocnemius compared to the NOR gastrocnemius. The positive correlations between inflammatory factors/MuRF1 and specific metabolites suggested potential metabolic alterations associated with inflammation and muscle atrophy. Conversely, the negative correlations with certain metabolites, such as ATP, 3-HB, and glutathione, highlighted their potential roles in mitigating inflammation and alleviating skeletal muscle atrophy. The observed opposite correlations involving phosphorylated p-AKT suggested its involvement in modulating these metabolic alterations. Overall, these results contributed to a better understanding of the complex interplay among inflammatory factors, protein expression, and metabolic changes in cancer cachexia.

### 3.7. Identification of Significantly Altered Metabolic Pathways

We identified significantly altered metabolic pathways based on the relative levels of metabolites from the three groups of the mouse gastrocnemius using two criteria: *p* < 0.05 and PIV > 0.1. In comparison to the NOR group, the CAC group exhibited eight significant pathways (Appendix A). Similarly, when comparing the CAC-K group with the CAC group, we identified seven significant pathways (Appendix A). It is important to note that although we identified several significantly altered metabolic pathways based on pairwise comparisons of the metabolite levels between the CAC and NOR gastrocnemius, as well as the CAC-K and CAC gastrocnemius, not all of these pathways may be primarily localized in skeletal muscle. Possibly, a few pathways are primarily present in other tissues, such as the liver, due to material exchange between skeletal muscle and the rest of the organism. These pathways likely reflected the global metabolic changes occurring in the organism and were closely associated with the metabolic changes observed in the cachectic gastrocnemius.

## 4. Discussion

Cachexia significantly impacts the survival rate and quality of life in cancer patients, emphasizing the urgent need for effective therapeutic strategies [1,5]. While the ketogenic diet shows promise in cancer treatment, its composition presents restrictions that hinder its extensive application for treating cancer cachexia [19,22]. As a commonly used food additive [30], EHB has the potential to overcome these restrictions. In this study, we evaluated the impact of EHB administration on 3-HB levels in both blood and skeletal muscle, then investigated the protective effects of EHB administration on cachectic muscle atrophy, and further clarified the underlying metabolic mechanisms.

We investigated the therapeutic potential of EHB administration by evaluating its impact on 3-HB levels in both blood and skeletal muscle. Furthermore, we examined the effects of EHB administration on cachectic muscle atrophy. Our results revealed that a single administration of EHB significantly increased the serum and gastrocnemius 3-HB levels, despite not detecting intact EHB in the samples (Figure 1). This suggests that EHB administration may induce similar effects to direct 3-HB treatment. Similar observations were reported in previous studies, where a single ingestion of (R)-3-hydroxybutyl (R)-3-hydroxybutyrate (KE) resulted in a transient elevation of plasma 3-HB levels in athletes and healthy individuals [24,25,37], without detecting intact KE. These findings were consistent with our results on the dynamic changes in serum 3-HB levels over time following EHB administration. Notably, this study represented the first investigation of the kinetic profiles of 3-HB levels in both the blood and skeletal muscle of mice following EHB administration.

Our study provided compelling evidence for the efficacy of EHB administration in alleviating cachectic symptoms in a mouse model of colon cancer cachexia (Figure 2). Our results demonstrated that EHB administration not only partially restores tumor-free body weight and fat mass (Figure 2B–D) but also significantly decreases the mass of colon tumors (Figure 2E). It was previously demonstrated that EHB can reduce the proliferation of colonic crypt cells and potently suppress intestinal tumor growth [38]. This evidence showed that EHB treatment has anti-tumor effects in addition to anti-cachectic effects. Considering that cancer cachexia primarily arises from the presence of the tumor itself, and the atrophy of skeletal muscle is largely driven by various known and unknown molecules released by tumor tissues, it is expected that the anti-cachectic effect of EHB treatment is closely linked to its anti-tumor effect.

Metabolic homoeostasis dysregulation, including impairments in protein synthesis and proteolysis, glucose metabolism, amino acid metabolism, and so on, makes substantial and significant contributions to skeletal muscle atrophy [39,40]. The TCA cycle is a critical pathway for energy production in skeletal muscle. In CAC mice, the rapid proliferation of tumor cells consumes a significant amount of energy, leading to an energy shortage in vital organs and tissues [41]. Compared to the NOR gastrocnemius, the CAC gastrocnemius showed a notable decrease in the level of fumarate, an important intermediate of the TCA cycle (Figure 6), along with a significant increase in several amino acids including lysine, proline, and branched-chain amino acids (valine, leucine, and isoleucine), which can participate in the TCA cycle. These results suggested that the TCA cycle is impaired in the cachectic gastrocnemius compared to the normal control. Notably, EHB administration partially reversed the alterations in fumarate and these amino acids (Figure 6), indicating that EHB might improve the TCA cycle and TCA cycle anaplerosis. Interestingly, despite the observed promotion of the TCA cycle, no significant alteration in ATP production was observed (Appendix A). The result might be indicative of a more rational energy metabolism in cachectic skeletal muscle, which potentially reduces unnecessary protein degradation as well. These findings raised intriguing questions regarding the underlying mechanisms responsible for this discrepancy, highlighting the need for further investigation.

Pro-inflammatory and pro-cachectic cytokines, such as interleukin-1, TNF-α, IFN-γ, and the TGF-β family, are known to induce the expression of MuRF1 and increase oxidative stress in skeletal muscle. This leads to the promotion of the ubiquitin-mediated degradation of muscle proteins, contributing to muscle wasting [11,12]. In our study, we found that EHB administration has a multifaceted impact on the cachectic gastrocnemius. It not only profoundly decreased the levels of multiple inflammatory factors but also significantly increased the total antioxidant capacity (Figure 3). Additionally, EHB administration downregulated the expression of MuRF1, thereby reducing the ubiquitin-mediated degradation of muscle proteins and attenuating proteolysis in the cachectic gastrocnemius (Figure 4).

Compared with the NOR gastrocnemius, the CAC gastrocnemius showed a substantial decrease in the total antioxidant capacity (TAC, Figure 3) and a significant reduction in glutathione (GSH, Figure 7). Previous studies emphasized the critical role of reactive oxygen species (ROS) in the initiation, progression, and metastasis of tumors, particularly in the pathological process of cancer cachexia [5]. Therefore, the decreased TAC and GSH levels in the CAC gastrocnemius likely reflected an imbalance toward excessive ROS. Importantly, EHB administration markedly increased the levels of TAC, GSH, and glycine. These results indicated that EHB administration reduces inflammation and increases antioxidant capacity in cachectic skeletal muscle, which reduces proteolysis and promotes protein synthesis through the intracellular signaling pathways [10,11], thereby alleviating skeletal muscle atrophy. The observed changes in the inflammatory and antioxidant profiles of the cachectic gastrocnemius following EHB administration provided mechanistic insights into the protective effects of EHB administration on cachectic skeletal muscle. By reducing inflammation and enhancing antioxidant capacity, EHB administration created a more favorable environment for maintaining muscle and function. These findings highlighted the therapeutic potential of EHB administration as a promising intervention for managing the devastating effects of cancer cachexia on skeletal muscle.

In our study, we observed significant body weight loss, skeletal muscle atrophy, and decreased expression of the protein-synthesis-related p-AKT in the gastrocnemius muscle of CAC mice, indicating an imbalance between the protein synthesis and proteolysis in the cachectic gastrocnemius. Significantly, EHB administration increased\p-AKT expression and promoted protein synthesis, effectively maintaining the skeletal muscle weight (Figure 4) and diameter (Figure 4). Previous studies showed that the phenylalanine level in skeletal muscle is related to muscle protein synthesis and proteolysis [42]. In this study, we observed an increased level of phenylalanine in the cachectic gastrocnemius, whereas EHB administration significantly decreased its level.

In addition, the CAC gastrocnemius exhibited elevated levels of several amino acids, including branched-chain amino acids (leucine, isoleucine, and valine), lysine, phenylalanine, tyrosine, proline, inosine, and 2-methylglutarate. These changes in metabolite levels positively correlated with the expressions of inflammatory factors and MuRF1, while they negatively correlated with the change in p-AKT expression, as indicated by Pearson’s correlation analysis of CAC vs. NOR (Appendix A). Conversely, there were decreases in metabolites such as ATP, 3-HB, glutathione, fumarate, glutamine, and pyruvate in the CAC gastrocnemius compared to the normal control, which were associated with the increased expression of MuRF1 but were inversely related to p-AKT expression. These findings provided compelling evidence that the beneficial effects of EHB administration on muscle atrophy and metabolic dysregulation are mediated by restoring the balance between protein synthesis and proteolysis and improving the overall metabolic homeostasis in cachectic skeletal muscle. The modulation of key metabolite levels and protein-synthesis-related signaling pathways by EHB administration was instrumental in preserving muscle mass and functionality in the context of cancer cachexia. By elucidating the underlying mechanisms by which EHB administration promotes muscle health, our study highlighted its potential as a promising therapeutic approach for the management of muscle-related diseases.

## 5. Conclusions

In summary, our study provides compelling evidence supporting the protective effects of EHB administration in colon cancer cachexia and elucidates the underlying metabolic mechanisms. The efficacy of EHB administration in alleviating cachectic gastrocnemius atrophy can be attributed to three key mechanisms: the promotion of the TCA cycle and the attenuation of proteolysis, the promotion of protein synthesis and the improvement of metabolic homeostasis, and a reduction in inflammation and an enhancement of the antioxidant capacity. By targeting these mechanisms, EHB administration partially restores the delicate balance between protein synthesis and proteolysis, thus ameliorating skeletal muscle wasting in CAC mice. In addition, these findings hold significant implications for the development of EHB administration as a ketone supplementation approach to achieve nutritional ketosis without the necessity of strict dietary restrictions.

## Figures and Tables

**Figure 1 biomolecules-13-01330-f001:**
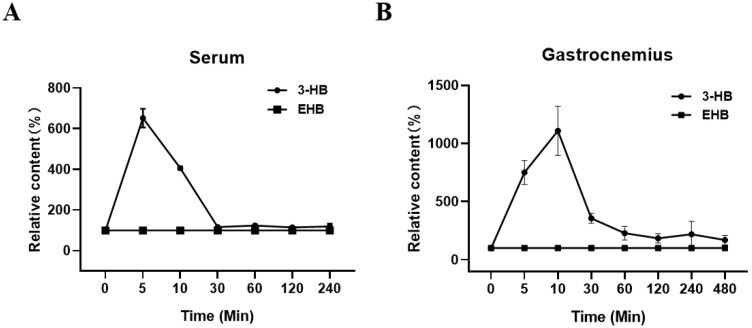
Relative levels of 3-HB and EHB in serum and gastrocnemius over time in normal mice following a single intraperitoneal injection of EHB. (**A**) Serum; (**B**) gastrocnemius. Integrals of 3-HB and EHB were measured from 1D ^1^H-NMR spectra recorded at 25 °C on a Bruker Avance III 850 MHz spectrometer. The integrals of 3-HB and EHB were normalized by those measured 0 min post-EHB administration to represent the relative 3-HB and EHB levels at different time points (n = 5 for each time point).

**Figure 2 biomolecules-13-01330-f002:**
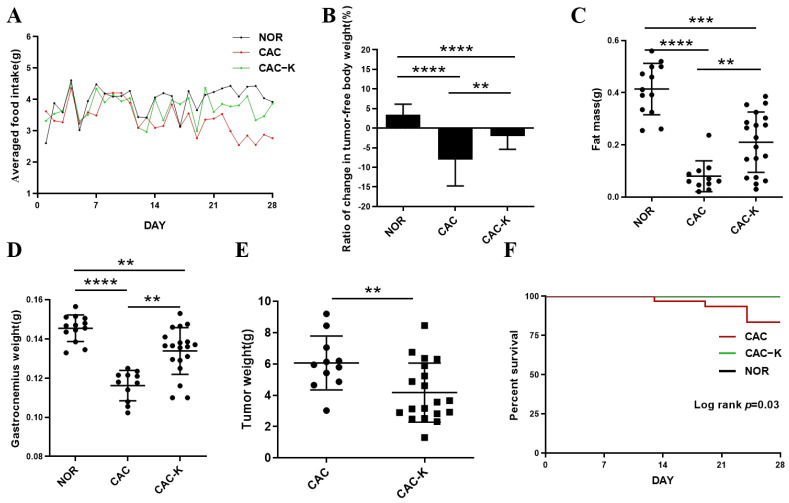
EHB administration alleviated cachectic symptoms in the mouse model of colon cancer cachexia. (**A**) Food intake curve; (**B**) ratio of change in tumor-free body weight (%); (**C**) epididymal fat mass; (**D**) gastrocnemius weight; (**E**) tumor weight; (**F**) survival curve. NOR, normal mice (n = 13); CAC, cachexia mice (n = 11); CAC-K, 3-HB-treated cachexia mice (n = 19). Experimental data are presented as mean ± SD. Statistical significances: *p* < 0.01, **; *p* < 0.001, ***; *p* < 0.0001, ****.

**Figure 3 biomolecules-13-01330-f003:**
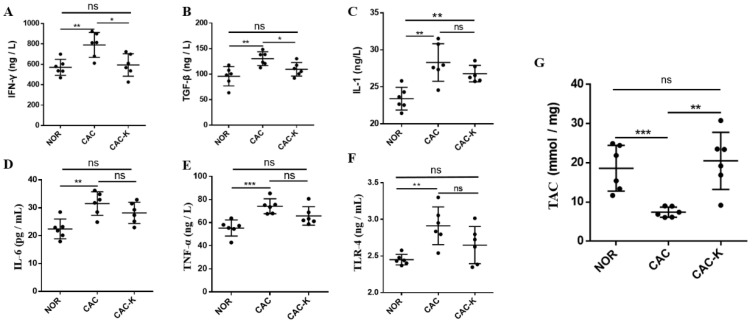
EHB administration decreased serum levels of inflammation factors and total antioxidant capacity in colon cancer cachexia mice. Pairwise comparisons between NOR and CAC and between CAC-K and CAC for serum levels of (**A**–**F**) inflammation factors including IFN-γ (**A**), TGF-β (**B**), IL-1 (**C**), IL-6 (**D**), TNF-α (**E**), TLR-4 (**F**), and (**G**) total antioxidant capacity (TAC) in gastrocnemius. Experimental data are reported as mean ± SD. Statistical significances: *p* > 0.05, ns; *p* < 0.05, *; *p* < 0.01, **; *p* < 0.001, ***.

**Figure 4 biomolecules-13-01330-f004:**
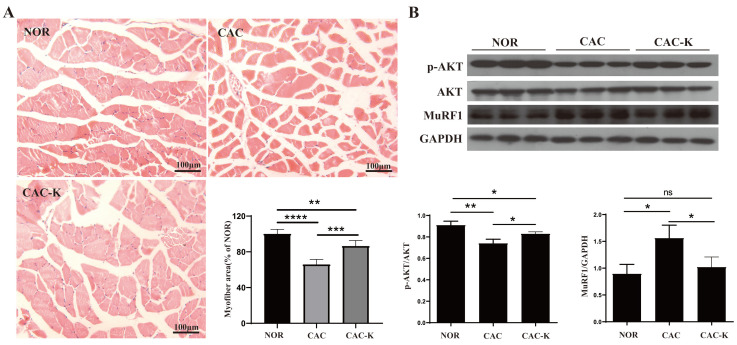
EHB-administration ameliorated skeletal muscle atrophy in the mouse model of colon cancer cachexia. (**A**) Hematoxylin–eosin staining in gastrocnemius muscle and quantitative comparisons of cross-sectional areas of skeletal muscle fibers for NOR vs. CAC and CAC-K vs. CAC (n = 6). (**B**) Expressions of p-AKT, AKT, MuRF1 in gastrocnemius, and pairwise comparisons of AKT phosphorylation and MuRF1 expression between NOR and CAC and between CAC-K and CAC (n = 3). Experimental data are reported as mean ± SD. Statistical significances: *p* > 0.05, ns; *p* < 0.05, *; *p* < 0.01, **; *p* < 0.001, ***; *p* < 0.0001, ****.

**Figure 5 biomolecules-13-01330-f005:**
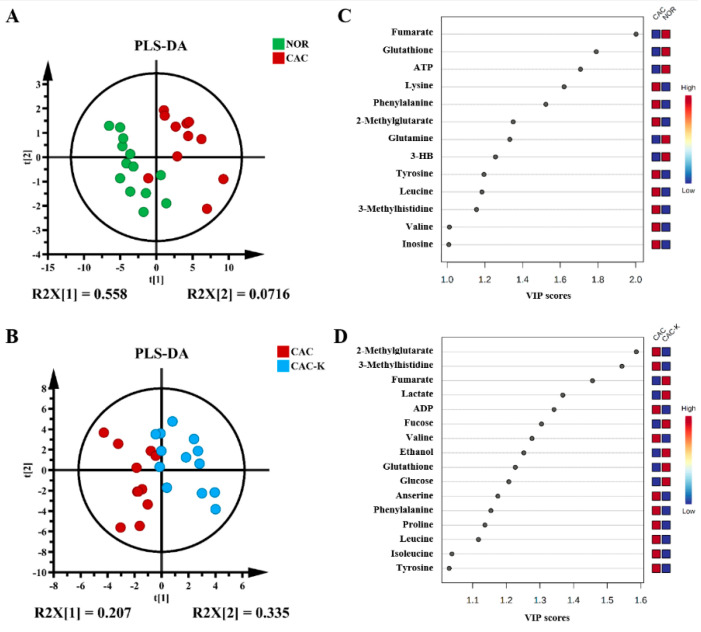
Multivariate statistical analysis for identifying significant metabolites from the comparisons of CAC vs. NOR and CAC-K vs. CAC. (**A**,**B**) Score plots of the PLS-DA models for CAC vs. NOR (**A**) and CAC-K vs. CAC (**B**). (**C**,**D**) VIP score-ranking plots of the significant metabolites identified by using the criterion of VIP > 1 calculated from the PLS-DA models of CAC vs. NOR (**C**) and CAC-K vs. CAC (**D**).

**Figure 6 biomolecules-13-01330-f006:**
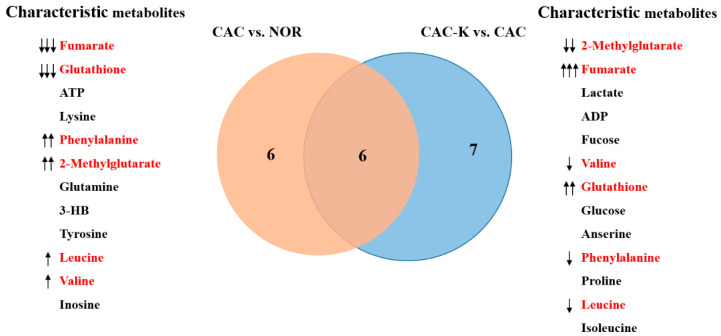
Venn diagram of the characteristic metabolites identified from pairwise comparisons of CAC vs. NOR and CAC-K vs. CAC. Characteristic metabolites were identified by using two criteria: VIP > 1 calculated from the PLS-DA model and *p* < 0.05 obtained from univariate analysis. Characterized metabolites highlighted in red were shared by pairwise comparisons. Note: ↑↑↑/↓↓↓, ↑↑/↓↓, and ↑/↓ represent that A was increased compared to B with a statistical significance of *p* < 0.05, *p* < 0.01, and *p* < 0.001, respectively.

**Figure 7 biomolecules-13-01330-f007:**
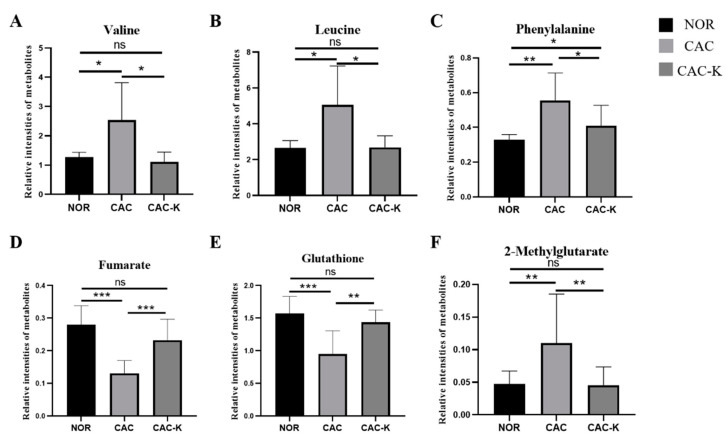
Quantitative comparisons of relative levels of the characterized metabolites shared by pairwise comparisons of CAC vs. NOR and CAC-K vs. CAC. (**A**) valine; (**B**) leucine; (**C**) phenylalanine; (**D**) fumarate; (**E**) glutathione; (**F**) 2-methylglutarate. Experimental data are represented as mean ± SD. Statistical significances: *p* > 0.05, ns; *p* <0.05, *; *p* < 0.01, **; *p* < 0.001, ***.

## Data Availability

Additional date are provided in the Appendix A.

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
