# Peer review of "Exploring the Therapeutic Potential of Ethyl 3-Hydroxybutyrate in Alleviating Skeletal Muscle Wasting in Cancer Cachexia"

_biomolecules, 2023, doi:10.3390/biom13091330_

Round 1

Reviewer 1 Report

Summary

This manuscript reports the effects of ethyl 3-hydroxybutyrate (EHB) in skeletal muscle wasting in the mouse model of cancer cachexia (CAC). Intraperitoneal injection of EHB and subsequent release of 3-hydroxybutyrate into the blood significantly ameliorated the cachectic phenotypes; body weight, fat mass, muscle weight, myofiber size, and secretion of inflammatory factors, etc. Metabonomic analysis revealed that EHB reversed the levels of some metabolites in CAC muscle to those in normal mice. This study provides the evidence of anti-cachectic effect of EHB, which will be useful in the therapy of cancer with cachexia. A few revisions and additional descriptions noted below will make the study perfect for publication in Biomolecules.

Comments

1.          There is no information on CT26 cell lines. It must be explained what kind of the cell line. Readers can not understand the experimental system.

2.          The doses of 3-HB (20 mM in vitro) (line 109) and EHB (300 mg/kg/day) (line 134) are generally very high. Have the authors tested other (lower) doses? Is there any other study using 3-HB or EHB at such high concentrations? The authors need to explain the reason for the concentration settings.

3.          In relation to the comment 1, were there any other phenotypes observed in the EHB-treated mice? Because EHB was used at a high dose, was released into the blood, and is a non-specific molecule, it may affect non-muscle tissues and/or non-cachectic phenomena. Anatomical observations, physiological properties (e.g., the shift in body weight, not the relative final weight in Figure 2B), and/or animal behaviors recorded by the authors should be described.

4.          Line 244 and Figure 2F: The manuscript says the survival rate of CAC-K mice was increased compared to that of CAC mice, but statistical significance was not shown. Statistical analysis must be performed, and the sentence should be revised if there was no significance.

5.          I basically agree with the authors’ hypothesis and discussion in lines 454-459; the increases of valine, leucine, and phenylalanine in gastrocnemius muscles of CAC mice (Figures 7A-7C) would be the results of impaired TCA cycle. Are there any references or previous studies that indicate elevations of these amino acids in the muscles of cachectic animals or patients?

6.          The current limitations of EHB for the human use need to be discussed. In particular, the volume, route (oral?), and duration of administration. Does EHB conflict with anti-cancer drugs? In relation to the comment 2, does EHB have a risk of side effects?

Minor points

7.          Line 16: “(EHB)” should be “EHB”.

8.          Line 16: “CAC mice” should be defined.

9.          Line 103: “fetal bovine serum” should be “FBS”.

10.          Line 144: “H&E staining” should be defined.

11.          Line 282: “we” should be “We”.

Author Response

Q1. There is no information on CT26 cell lines. It must be explained what kind of the cell line. Readers can not understand the experimental system.

A1. Thanks for your comment. We have provided more information about CT26 cell lines in the revised manuscript as follow:

“The mouse colon cancer cells (CT26) were obtained from National Biomedical Cell Resource Bank (BMCR, Beijing). ” (Line 92-93)

Q2. The doses of 3-HB (20 mM in vitro) (line 109) and EHB (300 mg/kg/day) (line 134) are generally very high. Have the authors tested other (lower) doses? Is there any other study using 3-HB or EHB at such high concentrations? The authors need to explain the reason for the concentration settings.

A2. Thank you for your constructive comment and suggestion. In this study, a pre-experiment was conducted to determine the doses of 3-HB (20 mM in vitro) and EHB (300 mg/kg/day) were determined in this study. Research has shown that ketone monoesters were used on mice with doses ranging from 300 to 750 (mg/kg)1-3, and exogenous 3-HB was administered on rats at a dose of 1000 (mg/ kg)4. Thus, the doses of 3-HB (20 mM in vitro) and EHB (300 mg/kg/day) may be within a reasonable range for mice.

References:

1.Yurista, S. R. et al. Ketone ester supplementation suppresses cardiac inflammation and improves cardiac energetics in a swine model of acute myocardial infarction. Metabolism 145, 155608, doi:10.1016/j.metabol.2023.155608 (2023).

2.Brady, A. J. & Egan, B. Acute Ingestion of a Ketone Monoester without Co-Ingestion of Carbohydrate Improves Running Economy in Male Endurance Runners. Medicine & Science in Sports & Exercise, doi:10.1249/mss.0000000000003278 (2023).

3.Falkenhain, K., Islam, H. & Little, J. P. Exogenous ketone supplementation: an emerging tool for physiologists with potential as a metabolic therapy. Exp Physiol 108, 177-187, doi:10.1113/Ep090430 (2023).

4.Horii, N. et al. Effect of Exogenous Acute β-Hydroxybutyrate Administration on Different Modalities of Exercise Performance in Healthy Rats. Medicine & Science in Sports & Exercise 55, 1184-1194, doi:10.1249/mss.0000000000003151 (2023).

Q3. In relation to the comment 1, were there any other phenotypes observed in the EHB-treated mice? Because EHB was used at a high dose, was released into the blood, and is a non-specific molecule, it may affect non-muscle tissues and/or non-cachectic phenomena. Anatomical observations, physiological properties (e.g., the shift in body weight, not the relative final weight in Figure 2B), and/or animal behaviors recorded by the authors should be described.

A3. Thank you for your constructive comment and suggestion. This study showed the beneficial effects of EHB in alleviating skeletal muscle wasting.  We observed that EHB treatment led to a decrease in the tumor weight of the mice in Figure 2E and a significant infiltration of inflammatory cells in the cachectic gastrocnemius muscle of colon cancer cachexia mice in Figure S2. The accepted diagnostic criteria for cachexia is known to be weight loss over 5%, so that is more suitable to use the ratio of weight changes (Fig 2B) as the index to diagnose and treatment of CAC1,2.

References:

  1. Martin, L. et al. Diagnostic criteria for cancer cachexia: reduced food intake and inflammation predict weight loss and survival in an international, multi-cohort analysis. J Cachexia Sarcopeni 12, 1189-1202, doi:10.1002/jcsm.12756 (2021).
  2. Arends, J. et al. Cancer cachexia in adult patients: ESMO Clinical Practice Guidelines. Esmo Open 6, doi:ARTN 10009210.1016/j.esmoop.2021.100092 (2021).

Q4.  Line 244 and Figure 2F: The manuscript says the survival rate of CAC-K mice was increased compared to that of CAC mice, but statistical significance was not shown. Statistical analysis must be performed, and the sentence should be revised if there was no significance.

A4. Thank you for your constructive comment and suggestion. We have conducted a statistical analysis to compare the survival rate of CAC-K mice to that of CAC mice, and the results are presented in Figure 2F, which show statistical significance.

 Figure 2. EHB administration alleviated cachectic symptoms in the mouse model of colon cancer cachexia. (F) Survival curve.

Q5. I basically agree with the authors’ hypothesis and discussion in lines 454-459; the increases of valine, leucine, and phenylalanine in gastrocnemius muscles of CAC mice (Figures 7A-7C) would be the results of impaired TCA cycle. Are there any references or previous studies that indicate elevations of these amino acids in the muscles of cachectic animals or patients?

A5. Thank you for your constructive comment. Skeletal muscle protein degradation and the release of amino acids are increased in cachexia. Previous studies have shown an altered metabolism of amino acids in cachectic muscle, with higher levels of these amino acids present in the muscles of cachectic animals or patients1-4

References:

  1. Viana, L. R. et al. Leucine-rich diet alters the H-1-NMR based metabolomic profile without changing the Walker-256 tumour mass in rats. Bmc Cancer 16, doi:ARTN 76410.1186/s12885-016-2811-2 (2016).
  2. Miyaguti, N. A. D. et al. Serum and Muscle(1)H NMR-Based Metabolomics Profiles Reveal Metabolic Changes Influenced by a Maternal Leucine-Rich Diet in Tumor-Bearing Adult Offspring Rats. Nutrients 12, doi:ARTN 210610.3390/nu12072106 (2020).
  3. Sirnio, P. et al. Alterations in serum amino-acid profile in the progression of colorectal cancer: associations with systemic inflammation, tumour stage and patient survival. Brit J Cancer 120, 238-246, doi:10.1038/s41416-018-0357-6 (2019).
  4. Cui, P. F. et al. Metabolic derangements of skeletal muscle from a murine model of glioma cachexia. Skeletal muscle 9, doi:ARTN 310.1186/s13395-018-0188-4 (2019).

Q6. The current limitations of EHB for the human use need to be discussed. In particular, the volume, route (oral?), and duration of administration. Does EHB conflict with anti-cancer drugs? In relation to the comment 2, does EHB have a risk of side effects?

A6. Thank you for your constructive comment. In this study, we were more interested in the therapeutic effect of our EHB on malignant skeletal muscle atrophy. Of course, the questions you raise are also crucial for the future use of EHB. Unfortunately, whether EHBs have conflicts with cancer drugs and other side effects has not been clarified in current research. There is insufficient data to support our discussion. We will continue to investigate these issues in future studies to better discuss them.

Q7. Minor points

  1. Line 16: “(EHB)” should be “EHB”.
  2. Line 16: “CAC mice” should be defined.
  3. Line 103: “fetal bovine serum” should be “FBS”.
  4. Line 144: “H&E staining” should be defined.
  5. Line 282: “we” should be “We”.

A7. Thanks for your constructive suggestions. We have made the following modifications in the revised manuscript.

  1. In line 16: We have changed (EHB)” to “EHB.
  2. In line 16: We have defined “CAC mice” as “cachectic mice”.
  3. Line 103: We have changed “fetal bovine serum” to “fetal bovine serum (FBS)”.
  4. Line 144: We have defined “H&E staining” as “the hematoxylin-eosin (H&E) staining”
  5. Line 282: We have changed “we” to “We”.

Reviewer 2 Report

 The aim of the study was to investigate the effects of ethyl 3-hydroxybutyrate administration (EHB)  on muscle atrophy in a mouse model of colon cancer cachexia.  It is concluded that EHB increases levels of 3-hydroxybutyrate (3-HB), ameliorates muscle atrophy, and improves survival. The authors suggest that the effects of EHB are mediated by 3-HB and that EHB administration enables to achieve nutritional ketosis without the need of dietary restriction.  The topic is of interest. The study appears to have been carefully conducted. 

 Comments:

1. Abstract - clarify the meaning of the CAC abbreviation.

2. Page 2, line 74 -  please, add information why EHB is used as a food additive.

 3. Can the mechanism of action of EHB (3-HB) be analogous to the action of beta-hydroxy-beta methylbutyrate (HMB), leucine metabolite with a proven protein anabolic effect?

4.  Methods – page 3, line 38. Why did the authors examine the effect of CAC and EHB on muscle wasting using the gastrocnemius muscle that is composed both by fast-twitch (glycolytic, type I) and slow-twitch (oxidative, type II) fibers? There are several articles (Biomolecules. 2020 Oct 23;10(11):1475; Physiol. Res. 2017 Dec 20;66(6):959-967; Int. J. Exp. Pathol. 2008 Feb;89(1):64-71; Amino Acids 2014;46(5):1377-1384) that show marked differences in the response to stress stimuli in slow-twitch muscles (e.g., soleus muscle) when compared with fast-twitch muscles (e.g., extensor digitorum longus and plantaris muscles).

5. Results -  data on EHB level are missing. Where does the conversion of EHB to 3-HB occur? Is it a spontaneous transformation or is an enzyme required?

6. Results – pages 6-7. Do the authors believe that changes in the level of "characteristic metabolites" in CAC animals treated by EHB are due to the beneficial effect of EHB (3-HB) on CAC? Aren't these rather metabolic changes induced by EHB administration? I lack a control group of intact animals treated with EHB.

6. Discussion

 - The authors should note that the effects of tumor and EHB on muscles may differ for fast and slow-twitch muscles.

- page 11, lines 417-419. Add reference.

- P. 12, lines 422 -426.  Avoid to repeat sentences of the same meaning and similar wording.

- P.12, lines 459 -466 and 482-492. Avoid unsubstantiated speculations.  The control group of healthy animals treated by EHB is necessary to prove whether the effect of EHB observed in animals with CAC is not only a metabolic response to EHB administration!

In summary, the authors must justify the design of the study and accordingly edit the Discussion section.

Author Response

Q1. Abstract - clarify the meaning of the CAC abbreviation.

A1. Thanks for your constructive suggestion. We have defined the CAC abbreviation as “cancer cachexia” in the revised Abstract.

Q2. Page 2, line 74 -  please, add information why EHB is used as a food additive.

A2. Thanks for your constructive suggestion. We have added the following information why EHB is used as a food additive in the revised manuscript (Line 71-73):

“Ethyl 3-hydroxybutyrate (EHB) is a ketone ester that is commonly used as a food additive due to its pleasant aroma and fruity flavour, which was originally discovered as a flavouring compound in wine [29,30].”

References:

  1. Jonfia-Essien, W. A., Alderson, P. G., Tucker, G., Linforth, R. & West, G. Behavioural responses of Tribolium castaneum (Herbst) to volatiles identified from dry cocoa beans. Pak J Biol Sci 10, 3549-3556, doi:10.3923/pjbs.2007.3549.3556 (2007).
  2. Ortega, C., Lopez, R., Cacho, J. & Ferreira, V. Fast analysis of important wine volatile compounds development and valida-tion of a new method based on gas chromatographic-flame ionisation detection analysis of dichloromethane microextracts. J Chromatogr A 923, 205-214, doi:10.1016/s0021-9673(01)00972-4 (2001).

Q3. Can the mechanism of action of EHB (3-HB) be analogous to the action of beta-hydroxy-beta methylbutyrate (HMB), leucine metabolite with a proven protein anabolic effect?

A3. Thanks for your constructive comment. Muscle atrophy is caused by a decrease in the rate of muscle protein synthesis (MPS) and/or an increase in the rate of muscle protein breakdown (MPB). This imbalance between MPS and MPB leads to a decrease in muscle mass. Mechanistically, this is thought to be due to the activation of catabolic pathways involving E3 ligases of the ubiquitin proteasome system, which increases MPB and/or reduces MPS through the inhibition of canonical mTOR signaling1. Studies have shown that HMB treatment can reduce skeletal muscle wasting by increasing MPS and decreasing MPB2-4, which is analogous to 3-HB (EHB) treatment. This can be thought to be due to the inhibition of ubiquitin–proteasome by HMB 5-6.

References:

  1. Gordon, B. S., Kelleher, A. R. & Kimball, S. R. Regulation of muscle protein synthesis and the effects of catabolic states. Int J Biochem Cell B 45, 2147-2157, doi:10.1016/j.biocel.2013.05.039 (2013).
  2. Prado, C. M., Orsso, C. E., Pereira, S. L., Atherton, P. J. & Deutz, N. E. P. Effects of beta-hydroxy beta-methylbutyrate (HMB) supplementation on muscle mass, function, and other outcomes in patients with cancer: a systematic review. J Cachexia Sarcopeni 13, 1623-1641, doi:10.1002/jcsm.12952 (2022).
  3. Wilkinson, D. J. et al. Effects of leucine and its metabolite -hydroxy--methylbutyrate on human skeletal muscle protein metabolism. J Physiol-London 591, 2911-2923, doi:10.1113/jphysiol.2013.253203 (2013).
  4. Eley, H. L., Russell, S. T., Baxter, J. H., Mukerji, P. & Tisdale, M. J. Signaling pathways initiated by beta-hydroxy-beta-methylbutyrate to attenuate the depression of protein synthesis in skeletal muscle in response to cachectic stimuli. Am J Physiol-Endoc M 293, E923-E931, doi:10.1152/ajpendo.00314.2007 (2007).
  5. Smith, H. J., Mukerji, P. & Tisdale, M. J. Attenuation of proteasome-induced proteolysis in skeletal muscle by beta-hydroxy-beta-methylbutyrate in cancer-induced muscle loss. Cancer Research 65, 277-283 (2005).
  6. Kovarik, M., Muthny, T., Sispera, L. & Holecek, M. Effects of beta-hydroxy-beta-methylbutyrate treatment in different types of skeletal muscle of intact and septic rats. Journal of Physiology and Biochemistry 66, 311-319, doi:10.1007/s13105-010-0037-3 (2010).

Q4. Methods – page 3, line 38. Wh composed bothy did the authors examine the effect of CAC and EHB on muscle wasting using the gastrocnemius muscle that is by fast-twitch (glycolytic, type I) and slow-twitch (oxidative, type II) fibers? There are several articles (Biomolecules. 2020 Oct 23;10(11):1475; Physiol. Res. 2017 Dec 20;66(6):959-967; Int. J. Exp. Pathol. 2008 Feb;89(1):64-71; Amino Acids 2014;46(5):1377-1384) that show marked differences in the response to stress stimuli in slow-twitch muscles (e.g., soleus muscle) when compared with fast-twitch muscles (e.g., extensor digitorum longus and plantaris muscles).

A4. Thank you for your constructive comment and suggestion. It is well known that cancer cachexia mainly induces atrophy of fast-twitch fibers, and the gastrocnemius muscle has a greater proportion of fast-twitch fibers than slow-twitch fibers. In addition, the gastrocnemius muscle is the largest flexor muscle in the leg. In mice, the gastrocnemius muscle is the only muscle large enough to be analyzed by NMR-based metabonomics, while other muscles (tibialis anterior and soleus, etc.) are too small. This study focused on the effects of EHB on the mouse gastrocnemius muscle (composed both by fast-twitch (glycolytic, type I) and slow-twitch (oxidative, type II) fibers), which is sufficient to demonstrate the role of EHB in alleviating skeletal muscle atrophy in cachexia1. We will continue to investigate the effects of EHB on different muscle types in the future.

  1. Lu, S. S. et al. Carnosol and its analogues attenuate muscle atrophy and fat lipolysis induced by cancer cachexia. J Cachexia Sarcopeni 12, 779-795, doi:10.1002/jcsm.12710 (2021).

Q5. Results -  data on EHB level are missing. Where does the conversion of EHB to 3-HB occur? Is it a spontaneous transformation or is an enzyme required?

A5. Thank you for your constructive comment and suggestion. We have added the data on EHB level in the revised Figure 1, and also added the sentence “However, no signal of EHB was present in serum and gastrocnemius over time in normal mice following a single intraperitoneal injection of EHB (Fig.1A and B).” in revised manuscript (Line 225-226).

Figure 1. Relative levels of 3-HB and EHB in serum and gastrocnemius over time in normal mice following a single intraperitoneal injection of EHB. Integrals of 3-HB and EHB were measured from 1D 1H-NMR spectra recorded at 25 ℃ on a Bruker Avance III 850 MHz spec-trometer. The integrals of 3-HB and EHB were normalized by those measured at 0 minutes post-EHB administration to represent the relative 3-HB and EHB levels at different time points (n=5 for each time point).

Our result (Not presented in this study) showing that EHB was present in the NMR spectrum recorded on the PBS sample, but was absent in the NMR spectra recorded on the mouse serum with and without in vitro EHB supplementation.  Note that the signal of 3-HB in the spectrum recorded on the serum with EHB supplementation was higher than that without EHB supplementation. This indicates that the transformation of EHB to 3-HB could happen in the serum, which was not a spontaneous transformation; instead, an unknown enzyme was required.

Figure. Amplifed resonance regions of 1D 1H-NMR spectra recorded separately on EHB dissolved in PBS buffer (red), mouse serum without EHB (blue), and mouse serum with EHB (green). The NMR spectra were recorded at 25℃ on a Bruker AVANCE III HD 850 MHz NMR spectrometer. No resonances of EHB were present in the spectra recorded on the serum.

Q6. Results – pages 6-7. Do the authors believe that changes in the level of "characteristic metabolites" in CAC animals treated by EHB are due to the beneficial effect of EHB (3-HB) on CAC? Aren't these rather metabolic changes induced by EHB administration? I lack a control group of intact animals treated with EHB.

A6. Thank you for your constructive comment and suggestion. The changes in the level of " characteristic metabolites " identified from CAC vs. CAC-K were induced by EHB administration. The changes in the level of "characteristic metabolites" identified from CON vs. CAC presents the metabolic response in cachexia mouse. however,the pairwise comparison of CAC vs. CAC-K shared six characteristic metabolites with that of CAC vs. NOR, with opposite changing trends (Figure 6). This suggests that EHB administration was able to significantly reverse the altered levels of these six characteristic metabolites in CAC gastrocnemius, effectively restoring them to normal levels (Figure 7). Therefore, it could be suggested that the changes in the levels of these "characteristic metabolites" shared by the two pairwise comparisons could be attributed to the beneficial effect of EHB (3-HB) on CAC.

Q7. Discussion

  1. The authors should note that the effects of tumor and EHB on muscles may differ for fast and slow-twitch muscles.
  2. page 11, lines 417-419. Add reference.
  3. page 12, lines 422 -426.  Avoid to repeat sentences of the same meaning and similar wording.
  4. page 12, lines 459-466 and 482-492. Avoid unsubstantiated speculations.  The control group of healthy animals treated by EHB is necessary to prove whether the effect of EHB observed in animals with CAC is not only a metabolic response to EHB administration!

A7. Thank you for your constructive comments and suggestions.

  1. We agree that the effects of tumor and EHB on muscles may differ for fast-twitch and slow-twitch muscles. Giving that cancer cachexia mainly induces atrophy of fast-twitch fibers, this study focused on the effect of EHB for alleviating atrophy of gastrocnemius muscle which had a greater proportion of fast-twitch fibers than slow-twitch fibers. We will continue to explore the effects of EHB on other fiber types in the future.
  2. We have added the following three references [19, 22, 37] in the revised manuscript (Line 417-420):

“While the ketogenic diet has shown promise in cancer treatment, its composition presents restrictions that hinder its extensive application for treating cancer cachexia [19,22]. As a commonly used food additive [37], EHB has the potential to overcome these restrictions.”

Reference:

  1. Lane, J., Brown, N. I., Williams, S., Plaisance, E. P. & Fontaine, K. R. Ketogenic Diet for Cancer: Critical Assessment and Research Recommendations. Nutrients 13, doi:10.3390/nu13103562 (2021).
  2. Yang, L. et al. Ketogenic diet and chemotherapy combine to disrupt pancreatic cancer metabolism and growth. Med 3, 119-136, doi:10.1016/j.medj.2021.12.008 (2022).
  3. Ortega, C., Lopez, R., Cacho, J. & Ferreira, V. Fast analysis of important wine volatile compounds Development and validation of a new method based on gas chromatographic-flame ionisation detection analysis of dichloromethane microextracts. Journal of Chromatography A 923, 205-214, doi:Doi 10.1016/S0021-9673(01)00972-4 (2001).
  4. We have reworded the content to avoid redundancy and repetition of similar phrases in the revised manuscript (Line 420-423).

‘In this study, we evaluated the impact of EHB administration on 3-HB levels in both blood and skeletal muscle, then investigated the protective effects of EHB administration on cachectic muscle atrophy and further clarified the underlying metabolic mechanisms.’

  1. As suggested in answer (A6) to question 6 (Q6), it could be suggested that the changes in the levels of these "characteristic metabolites" shared by the two pairwise comparisons could be attributed to the beneficial effect of EHB (3-HB) on CAC.

Round 2

Reviewer 2 Report

Unfortunately, I have to state that the authors did not make changes to the manuscript that would allow it to be published. Authors must add a control group of healthy animals treated by EHB that would prove that metabolic alterations observed in EHB treated animals with CAC indicate positive therapeutic effects and are not only a response to EHB administration observed also in healthy subjects. Without this control, there is a risk of justified criticism of the entire work.